# R-GAP: Recursive Gradient Attack on Privacy

Junyi Zhu and Matthew Blaschko

Dept. ESAT, Center for Processing Speech and Images
KU Leuven, Belgium
{junyi.zhu,matthew.blaschko}@esat.kuleuven.be

## Abstract

Federated learning frameworks have been regarded as a promising approach to break the dilemma between demands on privacy and the promise of learning from large collections of distributed data. Many such frameworks only ask collaborators to share their local update of a common model, i.e. gradients, instead of exposing their raw data to other collaborators. However, recent optimization-based gradient attacks show that raw data can often be accurately recovered from gradients. It has been shown that minimizing the Euclidean distance between true gradients and those calculated from estimated data is often effective in fully recovering private data. However, there is a fundamental lack of theoretical understanding of how and when gradients can lead to unique recovery of original data. Our research fills this gap by providing a closed-form recursive procedure to recover data from gradients in deep neural networks. We name it *Recursive Gradient Attack on Privacy* (R-GAP). Experimental results demonstrate that R-GAP works as well as or even better than optimization-based approaches at a fraction of the computation under certain conditions. Additionally, we propose a *Rank Analysis* method, which can be used to estimate the risk of gradient attacks inherent in certain network architectures, regardless of whether an optimization-based or closed-form-recursive attack is used. Experimental results demonstrate the utility of the rank analysis towards improving the network's security. Source code is available for download from https://github.com/JunyiZhu-AI/R-GAP.

## 1 Introduction

Distributed and federated learning have become common strategies for training neural networks without transferring data (Jochems et al., 2016; 2017; Konečný et al., 2016; McMahan et al., 2017). Instead, model updates, often in the form of gradients, are exchanged between participating nodes. These are then used to update at each node a copy of the model. This has been widely applied for privacy purposes (Rigaki & Garcia, 2020; Cristofaro, 2020), including with medical data (Jochems et al., 2016; 2017). Recently, it has been demonstrated that this family of approaches is susceptible to attacks that can in some circumstances recover the training data from the gradient information exchanged in such federated learning approaches, calling into question their suitability for privacy preserving distributed machine learning (Phong et al., 2018; Wang et al., 2019; Zhu et al., 2019; Zhao et al., 2020; Geiping et al., 2020; Wei et al., 2020). To date these attack strategies have broadly fallen into two groups: (i) an analytical attack based on the use of gradients with respect to a bias term (Phong et al., 2018), and (ii) an optimization-based attack (Zhu et al., 2019) that can in some circumstances recover individual training samples in a batch, but that involves a difficult non-convex optimization that doesn't always converge to a correct solution (Geiping et al., 2020), and that provides comparatively little insights into the information that is being exploited in the attack.

The development of privacy attacks is most important because they inform strategies for protecting against them. This is achieved by perturbations to the transferred gradients, and the form of the attack can give insights into the type of perturbation that can effectively protect the data (Fan et al., 2020). As such, the development of novel closed-form attacks is essential to the analysis of privacy in federated learning. More broadly, the existence of model inversion attacks (He et al., 2019; Wang et al., 2019; Yang et al., 2019; Zhang et al., 2020) calls into question whether transferring

a fully trained model can be considered privacy preserving. As the weights of a model trained by (stochastic) gradient descent are the summation of individual gradients, understanding gradient attacks can assist in the analysis of and protection against model inversion attacks in and outside of a federated learning setting.

In this work, we develop a novel third family of attacks, *recursive gradient attack on privacy* (R-GAP), that is based on a recursive, depth-wise algorithm for recovering training data from gradient information. Different from the analytical attack using the bias term, R-GAP utilizes much more information and is the first closed-form algorithm that works on both convolutional networks and fully connected networks with or without bias term. Compared to optimization-based attacks, it is not susceptible to local optima, and is orders of magnitude faster to run with a deterministic running time. Furthermore, we show that under certain conditions our recursive attack can fully recover training data in cases where optimization attacks fail. Additionally, the insights gained from the closed form of our recursive attack have lead to a refined *rank analysis* that predicts which network architectures enable full recovery, and which lead to provable noisy recovery due to rank-deficiency. This explains well the performance of both closed-form and optimization-based attacks. We also demonstrate that using rank analysis we are able to make small modifications to network architectures to increase the network's security without sacrificing its accuracy.

## 1.1 RELATED WORK

**Bias attacks**: The original discovery of the existence of an analytical attack based on gradients with respect to the bias term is due to Phong et al. (2018). Fan et al. (2020) also analyzed the bias attack as a system of linear equations, and proposed a method of perturbing the gradients to protect against it. Their work considers convolutional and fully-connected networks as equivalent, but this ignores the aggregation of gradients in convolutional networks. Similar to our work, they also perform a rank analysis, but it considers fewer constraints than is included in our analysis (Section 4).

**Optimization attacks**: The first attack that utilized an optimization approach to minimize the distance between gradients appears to be due to Wang et al. (2019). In this work, optimization is adopted as a submodule in their GAN-style framework. Subsequently, Zhu et al. (2019) proposed a method called *deep leakage from gradients* (DLG) which relies entirely on minimization of the difference of gradients (Section 2). They propose the use of L-BFGS (Liu & Nocedal, 1989) to perform the optimization. Zhao et al. (2020) further analyzed label inference in this setting, proposing an analytic way to reconstruct the one-hot label of multi-class classification in terms of a single input. Wei et al. (2020) show that DLG is sensitive to initialization and proposed that the same class image is an optimal initialization. They proposed to use SSIM as image similarity metric, which can then be used to guide optimization by DLG. Geiping et al. (2020) point out that as DLG requires second-order derivatives, L-BFGS actually requires third-order derivatives, which leads to challenging optimzation for networks with activation functions such as ReLU and LeakyReLU. They therefore propose to replace L-BFGS with Adam (Kingma & Ba, 2015). Similar to the work of Wei et al. (2020), Geiping et al. (2020) propose to incorporate an image prior, in this case total variation, while using PSNR as a quality measurement.

## 2 OPTIMIZATION-BASED GRADIENT ATTACKS ON PRIVACY (**O-GAP**)

Optimization-based gradient attacks on privacy (O-GAP) take the real gradients as its ground-truth label and utilizes optimization to decrease the distance between the real gradients $\nabla \mathbf{W}$ and the dummy gradients $\nabla \mathbf{W}'$ generated by a pair of randomly initialized dummy data and dummy label. The objective function of O-GAP can be generally expressed as:

$$\arg \min_{x', y'} \|\nabla \mathbf{W} - \nabla \mathbf{W}'\|^2 = \arg \min_{x', y'} \sum_{i=1}^{d} \|\nabla \mathbf{W}_i - \nabla \mathbf{W}'_i\|^2, \tag{1}$$

where the summation is taken over the layers of a network of depth $d$, and $(x', y')$ is the dummy training data and label used to generate $\nabla W'$. The idea of O-GAP was proposed by Wang et al. (2019). However, they have adopted it as a part of their GAN-style framework and did not realize that O-GAP is able to preform a more accurate attack by itself. Later in the work of Zhu et al. (2019), O-GAP has been proposed as a stand alone approach, the framework has been named as Deep Leakage from Gradients (DLG).

The approach is intuitively simple, and in practice has been shown to give surprisingly good results (Zhu et al., 2019). However, it is sensitive to initialization and prone to fail (Zhao et al., 2020). The choice of optimizer is therefore important, and convergence can be very slow (Geiping et al., 2020). Perhaps most importantly, Equation 1 gives little insight into what information in the gradients is being exploited to recover the data. Analysis in Zhu et al. (2019) is limited to empirical insights, and fundamental open questions remain: *What are sufficient conditions for* $\arg\min_{x',y'} \sum_{i=1}^{d} \|\nabla W_i - \nabla W_i'\|^2$ *to have a unique minimizer?* We address this question in Section 4, and subsequently validate our findings empirically.

## 3    CLOSED-FORM GRADIENT ATTACKS ON PRIVACY

The first attempt of closed-form GAP was proposed in a research of privacy-preserving deep learning by Phong et al. (2018).

**Theorem 1** (Phong et al. (2018)). *Assume a layer of a fully connected network with a bias term, expressed as:*

$$Wx + b = z, \tag{2}$$

*where* $W, b$ *denote the weight matrix and bias vector, and* $x, z$ *denote the input vector and output vector of this layer. If the loss function* $\ell$ *of the network can be expressed as:*

$$\ell = \ell(f(x), y)$$

*where* $f$ *indicates a nested function of* $x$ *including activation function and all subsequent layers,* $y$ *is the ground-truth label. Then* $x$ *can be derived from gradients w.r.t.* $W$ *and gradients w.r.t.* $b$, *i.e.:*

$$\frac{\partial \ell}{\partial W} = \frac{\partial \ell}{\partial z} x^\top, \quad \frac{\partial \ell}{\partial b} = \frac{\partial \ell}{\partial z}$$

$$x^\top = \frac{\partial \ell}{\partial W_j} \Big/ \frac{\partial \ell}{\partial b_j} \tag{3}$$

*where* $j$ *denotes the* $j$-*th row, note that in fact from each row we can compute a copy of* $x^\top$.

When this layer is the first layer of a network, it is possible to reconstruct the data, i.e. $\mathbf{x}$, using this approach. In the case of noisy gradients, we can make use of the redundancy in estimating $\mathbf{x}$ by averaging over noisy estimates: $\hat{\mathbf{x}}^\top = \sum_j \frac{\partial \ell}{\partial W_j} \big/ \frac{\partial \ell}{\partial b_j}$. However, simply removing the bias term can disable this attack. Besides, this approach does not work on convolutional neural networks due to a dimension mismatch in Equation 3. Both of these two problems have been resolved in our approach.

### 3.1    RECURSIVE GRADIENT ATTACK ON PRIVACY (**R-GAP**)

For simplicity we derive the R-GAP in terms of binary classification with a single image as input. In this setting we can generally describe the network and loss function as:

$$\mu = y \mathbf{w}_d \, \sigma_{d-1} \overbrace{\left( \mathbf{W}_{d-1} \underbrace{\sigma_{d-2} \left( \mathbf{W}_{d-2} \phi \left( \mathbf{x} \right) \right)}_{=: f_{d-2}(\mathbf{x})} \right)}^{=: f_{d-1}(\mathbf{x})} \tag{4}$$

$$\ell = \log(1 + e^{-\mu}) \tag{5}$$

where $y \in \{-1, 1\}$, $d$ denotes the $d$-th layer, $\phi$ represents all layers previous to $d-2$, and $\sigma$ denotes the activation function. Note that, although our notation omits the bias term in our approach, with an augmented matrix and augmented vector it is able to represent both of the linear map and the translation, e.g. Equation 2, using matrix multiplication as shown in Equation 4. So our formulation also includes the approach proposed by Phong et al. (2018). Moreover, if the $i$-th layer is a convolutional layer, then $\mathbf{W}_i$ is an extended circulant matrix representing the convolutional kernel (Golub & Van Loan, 1996), and data $\mathbf{x}$ as well as input of each layer are represented by a flattened vector in Equation 4.

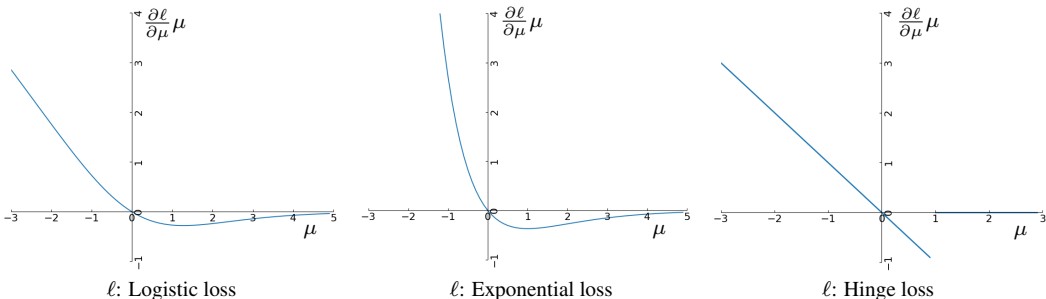

Figure 1: In consideration of logistic loss, exponential loss and hinge loss, $\frac{\partial \ell}{\partial \mu}\mu$ is not monotonic w.r.t. $\mu$. It is equal to 0 at $\mu = 0$, after that it either approximates $0^-$, or equals to 0 after decreasing to $\mu = 1$.

### 3.1.1 RECOVERING DATA FROM GRADIENTS

From Equation 4 and Equation 5 we can derive following gradients:

$$\frac{\partial \ell}{\partial \mathbf{w}_d} = y \frac{\partial \ell}{\partial \mu} f_{d-1}^\top \tag{6}$$

$$\frac{\partial \ell}{\partial \mathbf{W}_{d-1}} = \left( \left( \mathbf{w}_d^\top \left( y \frac{\partial \ell}{\partial \mu} \right) \right) \odot \sigma_{d-1}' \right) f_{d-2}^\top \tag{7}$$

$$\frac{\partial \ell}{\partial \mathbf{W}_{d-2}} = \left( \left( \mathbf{W}_{d-1}^\top \left( \left( \mathbf{w}_d^\top \left( y \frac{\partial \ell}{\partial \mu} \right) \right) \odot \sigma_{d-1}' \right) \right) \odot \sigma_{d-2}' \right) \phi^\top \tag{8}$$

where $\sigma'$ denotes the derivative of $\sigma$, for more details of deriving the gradients refer to Appendix H. The first observation of these gradients is that:

$$\frac{\partial \ell}{\partial \mathbf{w}_d} \cdot \mathbf{w}_d = \frac{\partial \ell}{\partial \mu} \mu \tag{9}$$

Additionally, if $\sigma_1, \ldots, \sigma_{d-1}$ are ReLU or LeakyRelu, the dot product of the gradients and weights of each layer will be the same, i.e.:

$$\frac{\partial \ell}{\partial \mathbf{w}_d} \cdot \mathbf{w}_d = \frac{\partial \ell}{\partial \mathbf{W}_{d-1}} \cdot \mathbf{W}_{d-1} = \ldots = \frac{\partial \ell}{\partial \mathbf{W}_1} \cdot \mathbf{W}_1 = \frac{\partial \ell}{\partial \mu} \mu \tag{10}$$

Since gradients and weights of each layer are known, we can obtain $\frac{\partial \ell}{\partial \mu}\mu$. If loss function $\ell$ is logistic loss (Equation 5), we obtain:

$$\frac{\partial \ell}{\partial \mu} \mu = \frac{-\mu}{1 + e^\mu}. \tag{11}$$

In order to perform R-GAP, we need to derive $\mu$ from $\frac{\partial \ell}{\partial \mu}\mu$. As we can see, $\frac{\partial \ell}{\partial \mu}\mu$ is non-monotonic, which means knowing $\frac{\partial \ell}{\partial \mu}\mu$ does not always allow us to uniquely recover $\mu$. However, even in the case that we cannot uniquely recover $\mu$, there are only two possible values to consider. Figure 1 illustrates $\frac{\partial \ell}{\partial \mu}\mu$ of logistic, exponential, and hinge losses, showing when we can uniquely recover $\mu$ from $\frac{\partial \ell}{\partial \mu}\mu$. The non-uniqueness of $\mu$ inspires us to find a sort of data that can trigger exactly the same gradients as the real data, which we name *twin data*, denoted by $\tilde{\mathbf{x}}$. The existence of twin data demonstrates that the objective function of DLG could have more than one global minimum, which explains at least in part why DLG is sensitive to initialization, for more information and experiments about the twin data refer to Appendix B.

The second observation on Equations 6-8 is that the gradients of each layer have a repeated format:

$$\frac{\partial \ell}{\partial \mathbf{w}_d} = \mathbf{k}_d f_{d-1}^\top; \ \mathbf{k}_d := y \frac{\partial \ell}{\partial \mu} \tag{12}$$

$$\frac{\partial \ell}{\partial \mathbf{W}_{d-1}} = \mathbf{k}_{d-1} f_{d-2}^\top; \ \mathbf{k}_{d-1} := \left( \mathbf{w}_d^\top \mathbf{k}_d \right) \odot \sigma_{d-1}' \tag{13}$$

$$\frac{\partial \ell}{\partial \mathbf{W}_{d-2}} = \mathbf{k}_{d-2} \phi^\top; \ \mathbf{k}_{d-2} := \left( \mathbf{W}_{d-1}^\top \mathbf{k}_{d-1} \right) \odot \sigma_{d-2}' \tag{14}$$

In Equation 12, the value of $y$ can be derived from the sign of the gradients at this layer if the activation function of previous layer is ReLU or Sigmoid, i.e. $f_{d-1} > 0$. For multi-class classification, $y$ can always be analytically derived as proved by Zhao et al. (2020). From Equations 12-14 we can see that gradients are actually linear constraints on the output of the previous layer, also the input of the current layer. We name these *gradient constraints*, which can be generally described as:

$$\mathbf{K}_i \mathbf{x}_i = \text{flatten}(\frac{\partial \ell}{\partial \mathbf{W}_i}), \tag{15}$$

where $i$ denotes $i$-th layer, $\mathbf{x}_i$ denotes the input and $\mathbf{K}_i$ is a coefficient matrix containing all gradient constraints at the $i$-th layer.

### 3.1.2 IMPLEMENTATION OF R-GAP

To reconstruct the input $\mathbf{x}_i$ from the gradients $\frac{\partial \ell}{\partial \mathbf{W}_i}$ at the $i$-th layer, we need to determine $\mathbf{K}_i$ or $\mathbf{k}_i$. The coefficient vector $\mathbf{k}_i$ solely relies on the reconstruction of the subsequent layer. For example in Equation 13, $\mathbf{k}_{d-1}$ consists of $\mathbf{w}_d, \mathbf{k}_d, \sigma'_{d-1}$, where $\mathbf{w}_d$ is known, and $\mathbf{k}_d$ and $\sigma'_{d-1}$ are products of the reconstruction at the $d$-th layer. More specifically, $\mathbf{k}_d$ can be calculated by deriving $y$ and $\mu$ as described in Section 3.1.1, $\sigma'_{d-1}$ can be derived from the reconstructed $f_{d-1}$. The condition for recovering $\mathbf{x}_i$ under gradient constraints $\mathbf{k}_i$ is that the rank of the coefficient matrix equals the number of entries of the input, $\text{rank}(\mathbf{K}_i) = |\mathbf{x}_i|$. Furthermore, if this rank condition holds for $i = 1, ..., d$, we are able to reconstruct the input at each layer and do this recursively back to the input of the first layer.

The number of gradient constraints is the same as the number of weights, i.e. $\text{rows}(\mathbf{K}_i) = |\mathbf{W}_i|$; $i = 1, ..., d$. Specifically, in the case of a fully connected layer we always have $\text{rank}(\mathbf{K}_i) = |\mathbf{x}_i|$, which implies *the reconstruction over FCNs is always feasible*. However in the case of a convolutional layer the matrix could possibly be rank-deficient to derive $\mathbf{x}$. Fortunately, from the view of recursive reconstruction and assuming we know the input of the subsequent layer, i.e. the output of the current layer, there is a new group of linear constraints which we name *weight constraints*:

$$\mathbf{W}_i \mathbf{x}_i = \mathbf{z}_i; \quad \mathbf{z}_i \leftarrow f_i \tag{16}$$

For a convolution layer, the $\mathbf{W}_i$ we use in this paper is the corresponding circulant matrix representing the convolutional kernel (Golub & Van Loan, 1996), so we can express the convolution in the form of Equation 16. In order to derive $\mathbf{z}_i$ from $f_i$, the activation function $\sigma_i$ should be monotonic. Commonly used activation functions satisfy this requirement. Note that for the ReLU activation function, a 0 value in $f_i$ will remove a constraint in $\mathbf{W}_i$. Otherwise, the number of weights constraints is equal to the number of entries in output, i.e. $\text{rows}(\mathbf{W}_i) = |\mathbf{z}_i|$; $i = 1, ..., d$. In CNNs the number of weight constraints $|\mathbf{z}_i|$ is much larger than the number of gradient constraints $|\mathbf{W}_i|$ in bottom layers, and well compensate for the lack of gradient constraints in those layers. It is worth noting that, due to the transformation from a CNN to a FCN using the circulant matrix, a CNN has been regarded equivalent to a FCN in the parallel work of Fan et al. (2020). However, we would like to point out that in consideration of the gradients w.r.t. the circulant matrix, what we obtain from a CNN are the aggregated gradients. Therefore, the number of valid gradient constraints in a CNN are much smaller than its corresponding FCN. Therefore, the conclusion of a rank analysis derived from a FCN cannot be directly applied to a CNN.

Moreover, padding in the $i$-th convolutional layer increases $|\mathbf{x}_i|$, but also involves the same number of constraints, so we omit this detail in the subsequent discussion. However, we have incorporated the corresponding constraints in our approach. Based on gradient constraints and weight constraints, we break the gradient attacks down to a recursive process of solving systems of linear equations, which we name R-GAP . The approach is detailed in Algorithm 1.

## 4 RANK ANALYSIS

For optimization-based gradient attacks such as DLG, it is hard to estimate whether it will converge to a unique solution given a network's architecture other than performing an empirical test. An intuitive assumption would be that the more parameters in the model, the greater the chance of unique recovery, since there will be more terms in the objective function constraining the solution. We provide here an analytic approach, with which it is easy to estimate the feasibility of performing

---

**Algorithm 1:** R-GAP (Notation is consistent with Equation 6 to Equation 15)

---

**Data:** i: i-th layer; $\mathbf{W}_i$: weights; $\nabla \mathbf{W}_i$: gradients;
**Result:** $\mathbf{x}_1$
**for** $i \leftarrow d$ *to 1* **do**
    **if** $i = d$ **then**
        $\frac{\partial \ell}{\partial \mu} \mu = \nabla \mathbf{W}_i \cdot \mathbf{W}_i$;
        $\mu \leftarrow \frac{\partial \ell}{\partial \mu} \mu; \mathbf{k}_i := y \frac{\partial \ell}{\partial \mu}; \mathbf{z}_i := \frac{\mu}{y}$;
    **else**
        /* Derive $\sigma'_i$ and $z_i$ from $f_i$. Note that $\mathbf{x}_{i+1} = f_i$.          */
        $\sigma'_i \leftarrow \mathbf{x}_{i+1}; \mathbf{z}_i \leftarrow \mathbf{x}_{i+1}$;
        $\mathbf{k}_i := (\mathbf{W}_{i+1}^\top \mathbf{k}_{i+1}) \odot \sigma'_i$;
    **end**
    $\mathbf{K}_i \leftarrow \mathbf{k}_i; \nabla \mathbf{w}_i := \text{flatten}(\nabla \mathbf{W}_i)$;
    $\mathbf{A}_i := \begin{bmatrix} \mathbf{W}_i \\ \mathbf{K}_i \end{bmatrix}; \mathbf{b}_i := \begin{bmatrix} \mathbf{z}_i \\ \nabla \mathbf{w}_i \end{bmatrix}$;
    $\mathbf{x}_i := \mathbf{A}_i^\dagger \mathbf{b}_i$ // $\mathbf{A}_i^\dagger$:Moore-Penrose pseudoinverse
**end**

---

the recursive gradient attack, which in turn is a good proxy to estimate when DLG converges to a good solution (see Figure 2).

Since R-GAP solves a sequence of linear equations, it is infeasible when the number of unknown parameters is more than the number of constraints at any $i$-th layer, i.e. $|\mathbf{x}_i| - |\mathbf{W}_i| - |\mathbf{z}_i| > 0$. More precisely, R-GAP requires that the rank of $\mathbf{A}_i$, which consists of $\mathbf{W}_i$ and $\mathbf{K}_i$ as shown in Algorithm 1, is equal to the number of input entries $|\mathbf{x}_i|$. However, $\mathbf{A}_i \mathbf{x}_i = \mathbf{z}_i$ does not include all effective constraints over $\mathbf{x}_i$. Because $\mathbf{x}_i$ is unique to $\mathbf{z}_{i-1}$ or partly unique in terms of the ReLU activation function, any constraint over $\mathbf{z}_{i-1}$ will limit the possible value of $\mathbf{x}_i$. On that note, suppose $|\mathbf{x}_{i-1}| = m$, $|\mathbf{z}_{i-1}| = n$ and the weight constraints at the $i-1$ layer is overdetermined, i.e. $\mathbf{W}_{i-1}\mathbf{x}_{i-1} = \mathbf{z}_{i-1}$; $m < n$, $rank(\mathbf{W}_{i-1}) = m$. Without the loss of generality, let the first $m$ entries of $\mathbf{z}_{i-1}$ be linearly independent, the $m+1, \ldots, n$ entries of $\mathbf{z}_{i-1}$ can be expressed as linear combination of the first $m$ entries, i.e. $\mathbf{M}\mathbf{z}_{i-1}^{1, \ldots, m} = \mathbf{z}_{i-1}^{m+1, \ldots, n}$. In other words, if the previous layers are overdetermined by weight constraints, the subsequent layer will have additional constraints, not merely its local weight constraints and gradient constraints. Since this type of additional constraint is not derived from the parameters of the layer that under reconstruction, we name them *virtual constraints* denoted by $\mathcal{V}$. When the activation function is the identity function, the virtual constraints are linear and can be readily derived. For the derivative of the activation function not being a constant, the virtual constraints will become non-linear. For more details about deriving the virtual constraints, refer to Appendix C. Optimization based attacks such as DLG are iterative algorithms based on gradient descent, and are able to implicitly utilize the non-linear virtual constraints. Therefore to provide a comprehensive estimate of the data vulnerability under gradient attacks, we also have to count the number of virtual constraints. It is worth noticing that virtual constraints can be passed along through the linear equation systems chain, but only in one direction that is to the subsequent layers. Next, we will informally use $|\mathcal{V}_i|$ to denote the number of virtual constraints at the $i$-th layer, which can be approximated by $\sum_{n=1}^{i-1} max(|\mathbf{z}_n| - |\mathbf{x}_n|, 0) - max(|\mathbf{x}_n| - |\mathbf{z}_n| - |\mathbf{W}_n|, 0)$. For more details refer to Appendix C. In practice, the real number of such constraints is dependent on the data, current weights, and choice of activation function.

These three types of constraints, gradient, weight and virtual constraints, are effective for predicting the risk of gradient attack. To conclude, we propose that $|\mathbf{x}_i| - |\mathbf{W}_i| - |\mathbf{z}_i| - |\mathcal{V}_i|$ is a good index to estimate the feasibility of fully recovering the input using gradient attacks at the $i$-th layer. We denote this value *rank analysis index (RA-i)*. Particularly, $|\mathbf{x}_i| - |\mathbf{W}_i| - |\mathbf{z}_i| - |\mathcal{V}_i| > 0$ indicates it is not possible to perform a complete reconstruction of the input, and the larger this index is, the poorer the quality of reconstruction will be. If the constraints in a particular problem are linearly independent, $|\mathbf{x}_i| - |\mathbf{W}_i| - |\mathbf{z}_i| - |\mathcal{V}_i| < 0$ implies the ability to fully recover the input. The quality of reconstruction of data is well estimated by the maximal RA-i of all layers, as shown in Figure 2. In practice, the layers close to the data usually have smaller RA-i due to fewer virtual constraints.

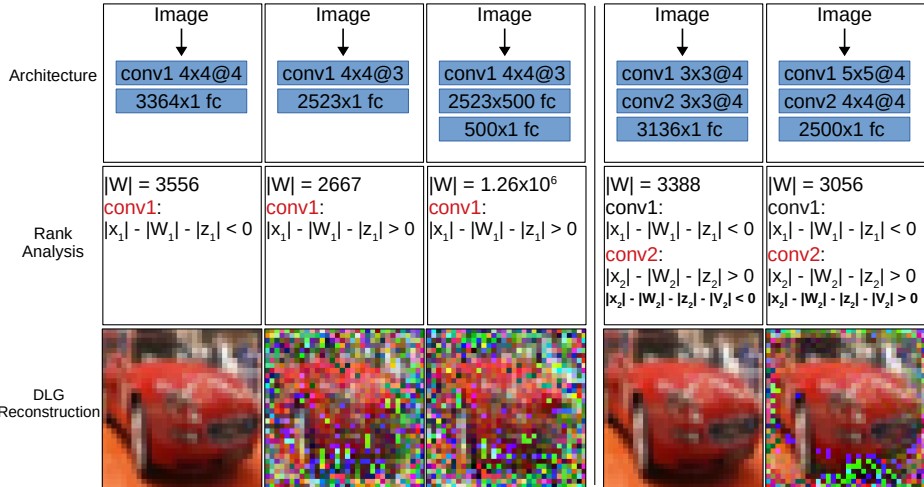

Figure 2: Estimating the privacy leakage of network through rank analysis. The critical layer for reconstruction has been red colored. First three columns show that even though bigger network has much more parameters denoted by $|W|$, which means we can collect more gradients to reconstruct the data, but if the layer close to data is rank-deficient, we are not able to fully recover the data. Despite that in the objective function of DLG, distance between all gradients will be reduced at the same time, redundant constraints in subsequent layer certainly cannot compensate the lack of constraints in previous layer. The fourth column shows that if rank-deficiency happens at the intermediate layer, redundant weight constraints in previous layer, i.e. virtual constraints, is able to compensate the deficiency at the intermediate layer. If a layer is rank-deficient after taking virtual constraints into account, fully recovery is again not possible as shown in the fifth column. However, as the rank analysis index of last column is smaller than the one of the second and third column, the reconstruction at the fifth column has a better quality. This figure demonstrates that rank analysis can correctly estimate the feasibility of performing DLG, for statistic result refer to Appendix A.

On top of that we analyse the residual block in ResNet, which shows some interesting traits of the skip connection in terms of the rank-deficiency, for more details refer to Appendix D.

A valuable observation we obtain through the rank analysis is that *the architecture rather than the number of parameters is critical to gradient attacks*, as shown in Figure 2. This observation is not obvious from simply specifying the DLG optimization problem(see Equation 1). Furthermore, since the data vulnerability of a network depends on the layer with maximal RA-i, we can design rank-deficiency into the architecture to improve the security of a network (see Figure 4).

## 5 RESULTS

Our novel approach R-GAP successfully extends the analytic gradient attack (Phong et al., 2018) from attacking a FCN with bias terms to attacking FCNs and CNNs[1] with or without bias terms. To test its performance, we use a CNN6 network as shown in Figure 3, which is full-rank considering gradient constraints and weight constraints. Additionally, we report results using a CNN6-d network, which is rank-deficient without consideration of virtual constraints, in order to to fairly compare the performance of DLG and R-GAP. CNN6-d has a CNN6 backbone and just decreases the output channel of the second convolutional layer to 20. The activation function is a LeakyReLU except the last layer, which is a Sigmoid. We have randomly initialized the network, as DLG is prone to fail if the network is at a late stage of training (Geiping et al., 2020). Furthermore, as the label can be analytically recovered by R-GAP, we always provide DLG the ground-truth label and let it recover the image only. Therefore the experiment actually compares R-GAP with iDLG (Zhao et al., 2020). The experimental results show that, due to an analytic one-shot process, run-time of R-GAP is orders of magnitude shorter than DLG. Moreover, R-GAP can recover the data more accurately,

---

[1]via equivalence between convolution and multiplication with a (block) circulant matrix.

while optimization-based methods like DLG recover the data with artifacts, as shown in Figure 3. The statistical results in Table 1 also show that the reconstruction of R-GAP has a much lower MSE than DLG on the CNN6 network. However, as R-GAP only considers gradient constraints and weight constraints in the current implementation, it does not work well on the CNN6-d network. Nonetheless, we find that it is easy to assess the quality of reconstruction of gradient attack without knowing the original image. As the better reconstruction has less salt-and-pepper type noise. We measure this by the difference of the image and its smoothed version (achieved by a simple 3x3 averaging) and select the output with the smaller norm. This hybrid approach which we name H-GAP combines the strengths of R-GAP and DLG, and obtains the best results.

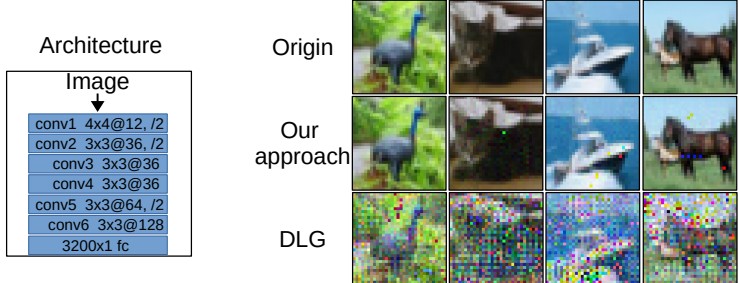

Figure 3: Performance of our approach and DLG over a CNN6 architecture. The diagram on the left demonstrates the network architecture on which we perform attack. The activation functions are LeakyReLU, except the last one which is Sigmoid.

|  | CNN6* | CNN6-d* | CNN6** | CNN6-d** |
|---|---|---|---|---|
| R-GAP | $0.010 \pm 0.0017$ | $1.4 \pm 0.073$ | $1.9 \times 10^{-4} \pm 7.0 \times 10^{-5}$ | $0.0090 \pm 9.3 \times 10^{-4}$ |
| DLG | $0.050 \pm 0.0014$ | $\mathbf{0.053 \pm 0.0016}$ | $4.2 \times 10^{-4} \pm 5.9 \times 10^{-5}$ | $\mathbf{0.0012 \pm 1.8 \times 10^{-4}}$ |
| H-GAP | $\mathbf{0.0069 \pm 0.0012}$ | $\mathbf{0.053 \pm 0.0016}$ | $\mathbf{1.4 \times 10^{-4} \pm 2.3 \times 10^{-5}}$ | $\mathbf{0.0012 \pm 1.8 \times 10^{-4}}$ |

*:CIFAR10    **:MNIST

Table 1: Comparison of the performance of R-GAP, DLG and H-GAP. MSE has been used to measure the quality of the reconstruction.

Moreover, we compare R-GAP with DLG on LeNet which has been benchmarked in DLG(Zhu et al., 2019), the statistical results are shown in Table 2. Both DLG and R-GAP perform well on LeNet. Empirically, if the MSE is around or below $1 \times 10^{-4}$, the difference of the reconstruction will be visually undetectable. However, we surprisingly find that by replacing the Sigmoid function with the Leaky ReLU, the reconstruction of DLG becomes much poorer. The condition number of matrix $\mathbf{A}$ (from Algorithm 1) changes significantly in this case. Since the Sigmoid function leads to a higher condition number at each convolutional layer, reconstruction error in the subsequent layer could be amplified in the previous layer, therefore DLG is forced to converge to a better result. In contrast, R-GAP has an accumulated error and naturally performs much better on LeNet*. Additionally, we find R-GAP could be a good initialization tool for DLG. As shown in the last column of Table 2, by initializing DLG with the reconstruction of R-GAP, and running 8% of the previous iterations, we achieve a visually indistinguishable result. However, for LeNet*, we find that DLG reduces the reconstruction quality obtained by R-GAP, which further shows the instability of DLG.

Our rank analysis is a useful offline tool to understand the risk inherent in certain network architectures. More precisely, we can use the rank analysis to find out the critical layer for the success of

|  | Condition number | | | MSE | | |
|---|---|---|---|---|---|---|
|  | conv1 | conv2 | conv3 | DLG | R-GAP | R-GAP$\rightarrow$ DLG |
| LeNet | $1.8 \times 10^4$ | $6.1 \times 10^3$ | $32.4$ | $3.7 \times 10^{-8}$ | $1.1 \times 10^{-4}$ | $1.1 \times 10^{-6}$ |
|  | $\pm 2.9$ | $\pm 0.3$ | $\pm 2.9 \times 10^{-4}$ | $\pm 8.6 \times 10^{-10}$ | $\pm 7.8 \times 10^{-6}$ | $\pm 1.1 \times 10^{-6}$ |
| LeNet* | $1.2 \times 10^3$ | $1.3 \times 10^3$ | $14.2$ | $5.2 \times 10^{-2}$ | $1.5 \times 10^{-10}$ | $4.8 \times 10^{-4}$ |
|  | $\pm 19.7$ | $\pm 22.5$ | $\pm 0.05$ | $\pm 2.9 \times 10^{-3}$ | $\pm 2.5 \times 10^{-11}$ | $\pm 9.1 \times 10^{-5}$ |

LeNet* is identical to LeNet but uses Leaky ReLU activation function instead of Sigmoid

Table 2: Comparison of R-GAP and DLG on LeNet benchmarked in DLG(Zhu et al., 2019).

gradient attacks and take precision measurements to improve the network's defendability. We report results on the ResNet-18, where the third residual block is critical since by cutting its skip connection the RA-i increases substantially. To perform the experiments, we use the approach proposed by Geiping et al. (2020), which extends DLG to incorporate image priors and performs better on deep networks. As shown in Figure 4, by cutting the skip connection of the third residual block, reconstructions become significantly poorer and more unstable. As a control test, cutting the skip connection of a non-critical residual block does not increase defendability noticeably. Note that two variants have the same or even slightly better performance on the classification task compared with the backbone. In previous works (Zhu et al., 2019; Wei et al., 2020), trade-off between accuracy and defendability of adding noise to gradients has been discussed. We show that using the rank analysis we are able to increase the defendability of a network with no cost in accuracy.

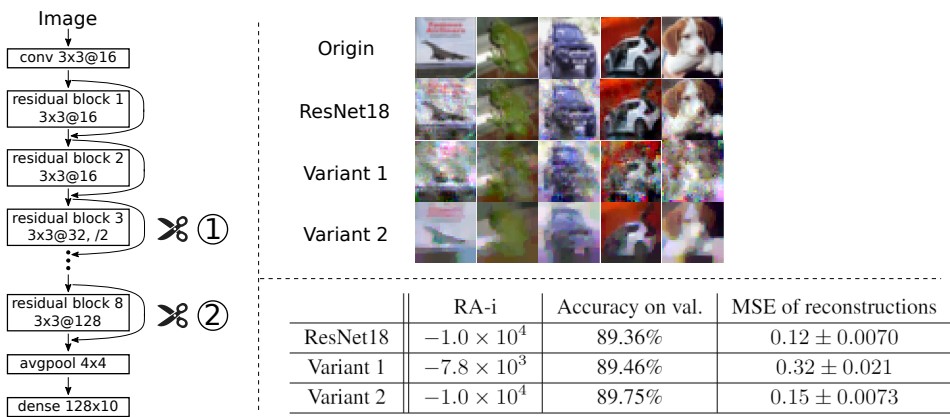

| | | RA-i | Accuracy on val. | MSE of reconstructions |
|---|---|---|---|---|
| ResNet18 | | $-1.0 \times 10^4$ | 89.36% | $0.12 \pm 0.0070$ |
| Variant 1 | | $-7.8 \times 10^3$ | 89.46% | $0.32 \pm 0.021$ |
| Variant 2 | | $-1.0 \times 10^4$ | 89.75% | $0.15 \pm 0.0073$ |

Figure 4: Left: Architectures of the ResNet18 with base width 16 and two variants. Variant 1 cuts the skip connection of the third residual block. Variant 2 cuts the skip connection of the eighth residual block. Upper right: Reconstruction examples of three networks. Lower right: Accuracy and reconstruction error of three networks. Training 200 epochs on CIFAR10 and saving the model with the best performance on the validation set, three networks achieve a close accuracy. Two variants perform even slightly better. In terms of gradient attacks, MSE of reconstructions from ResNet18 and Variant 2 are similar, since Variant 2 cut the skip connection of a non-critical layer and the RA-i does not change. Whereas, by cuting the skip connection of a critical layer, according to the rank analysis, increases RA-i substantially. MSE of the reconstructions from Variant 1 increases by nearly a factor of three with higher variance.

## 6 Discussion and conclusions

R-GAP makes the first step towards a general analytic gradient attack and provides a framework to answer questions about the functioning of optimization-based attacks. It also opens new questions, such as how to analytically reconstruct a minibatch of images, especially considering non-uniqueness due to permutation of the image indices. Nonetheless, we believe that by studying these questions, we can gain deeper insights into gradient attacks and privacy secure federated learning.

In this paper, we propose a novel approach R-GAP, which has achieved an analytic gradient attack for CNNs for the first time. Through analysing the recursive reconstruction process, we propose a novel rank analysis to estimate the feasibility of performing gradient based privacy attacks given a network architecture. Our rank analysis can be applied to the analysis of both closed-form and optimization-based attacks such as DLG. Using our rank analysis, we are able to determine network modifications that maximally improve the network's security, empirically without sacrificing its accuracy. Furthermore, we have analyzed the existence of twin data using R-GAP, which can explain at least in part why DLG is sensitive to initialization and what type of initialization is optimal. In summary, our work proposes a novel type of gradient attack, a risk estimation tool and advances the understanding of optimization-based gradient attacks.

ACKNOWLEDGEMENTS

This research received funding from the Flemish Government (AI Research Program).

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

## A    QUANTITATIVE RESULTS OF RANK ANALYSIS

A quantitative analysis of the predictive performance of the rank analysis index for the mean squared error of reconstruction is shown in Table 3.

| RA-i | -484 | 405 | 405 | -208 | 316 |
|---|---|---|---|---|---|
| MSE | $\mathbf{4.2 \times 10^{-9}}$ $\pm \mathbf{2.2 \times 10^{-9}}$ | $0.056 \pm 0.0035$ | $0.063 \pm 0.004$ | $\mathbf{2.7 \times 10^{-4}}$ $\pm \mathbf{2.0 \times 10^{-5}}$ | $0.013$ $\pm 7.2 \times 10^{-4}$ |

Table 3: Mean square error of the reconstruction over test set of CIFAR10. The corresponding network architecture has been shown in Figure 2 in the same order. Rank analysis index (RA-i) clearly predicts the reconstruction error. We can also regard RA-i as the security level of a network. A negative value indicates that the gradients of the network are able to fully expose the data, i.e. insecure, while a positive value indicates that completely recover the data from gradients is not possible. On top of that, higher RA-i indicate higher reconstruction error, therefore the network is more secure. According to our experiment, if the order of magnitude of MSE is equal to or less than $10^{-4}$, we could barely visually distinguish the recovered and real data, as shown in the fourth column of Figure 2. Note that, as the network gets deeper, DLG will become vulnerable, R-GAP will also be effected by numerical error. Besides that, DLG is sensitive to the initialization of dummy data, while R-GAP also needs to confirm the $\mu$ if it is not unique. Therefore, RA-i provides a reasonable upper bound of the privacy risk rather than quality prediction of one reconstruction.

## B    TWIN DATA

As we know $\frac{\partial \ell}{\partial \mu} \mu$ is non-monotonic as shown in Figure 1, which means knowing $\frac{\partial \ell}{\partial \mu} \mu$ does not always allow us to uniquely recover $\mu$. It is relatively straightforward to show that for monotonic convex losses (Bartlett et al., 2006), $\frac{\partial \ell}{\partial \mu} \mu$ is invertible for $\mu < 0$, $\frac{\partial \ell}{\partial \mu} \mu \leq 0$ for $\mu \geq 0$, and $\lim_{\mu \to \infty} \frac{\partial \ell}{\partial \mu} \mu = 0$. Due to the non-uniqueness of $\mu$ w.r.t to $\frac{\partial \ell}{\partial \mu} \mu$, we have:

$$\exists \, \mathbf{x}, \tilde{\mathbf{x}} \quad \text{s.t.} \quad \mu \neq \tilde{\mu}; \quad \frac{\partial \ell}{\partial \mu} \mu = \frac{\partial \ell}{\partial \tilde{\mu}} \tilde{\mu} \tag{17}$$

where $\mathbf{x}$ is the real data.

Taking the common setting that activation functions are ReLU or LeakyReLU, we can derive from Eq. 10 that:

$$\frac{\partial \ell}{\partial \mathbf{W}_i} \cdot \mathbf{W}_i = \frac{\partial \ell}{\partial \tilde{\mathbf{W}}} \cdot \tilde{\mathbf{W}}_i; \quad i = 1, \ldots, d \tag{18}$$

if there is a $\tilde{\mathbf{W}}_i$ is equal to $\mathbf{W}_i$, whereas the corresponding $\tilde{\mathbf{x}}$ is not same as $\mathbf{x}$ since $\mu \neq \tilde{\mu}$, we can find a data point that differs from the true data but leads to the same gradients. We name such data *twin data*, denoted by $\tilde{\mathbf{x}}$. As we know the gradients and $\mu$ of the twin data $\tilde{\mathbf{x}}$, by just giving them to R-GAP, we are able to easily find out the twin data. As shown in in Figure 5, twin data is actually proportional to the real data and smaller than it, which can also be straightforwardly derived from Equation 6 to Equation 8. Since the twin data and the real data trigger the same gradients, by decreasing the distance of gradients as Equation 1, DLG is suppose to converge to either of these data. As shown in Figure 5, we initialize DLG with a data close to the twin data $\tilde{\mathbf{x}}$, DLG converges to the twin data. In the work of Wei et al. (2020), the authors argue that using an image from the same

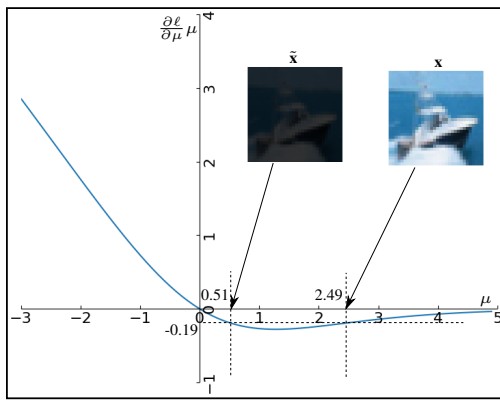 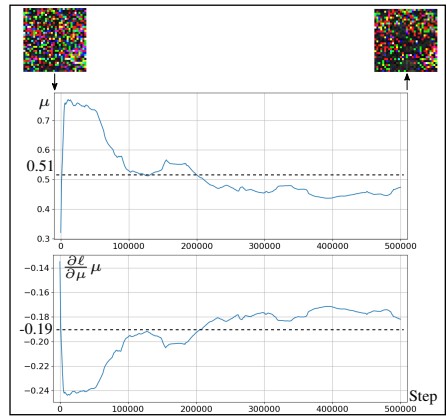

Figure 5: Twin data. The left figure demonstrates a twin data $\tilde{\mathbf{x}}$, which will trigger exactly the same gradients as the real data $\mathbf{x}$ does. Therefore, from the perspective of DLG, these two data are global minimum for the objective function. The right figure shows that by adding noise to shift the twin data a little and using it as an initialization, DLG will converge to the twin data rather than real data.

class as the real data would be the optimal initialization and empirically prove that. We want to point out that twin data is one important factor why DLG is so sensitive to the initialization and prone to fail with random initialization of dummy data particularly after some training steps of the network. Since DLG converges either to the twin data or the real data depends on the distance between these two data and the initialization, an image of the same class is usually close to the real data, therefore, DLG works better with that. While, with respect to $\mu$ or the prediction of the network, a random initialization is close to the twin data, so DLG converges to the twin data. However, the twin data has extremely small value, so any noise that comes up with optimization process stands out in the last result as shown in Figure 5.

It is worth noting that the twin data can be fully reconstructed only if RA-i $< 0$. In other words, if complete reconstruction is feasible and the twin data exits, R-GAP and DLG can recover either the twin data or real data depend on the initialization. But both of them lead to privacy leakage.

## C  VIRTUAL CONSTRAINTS

In this section we investigate the virtual constraints as proposed in the rank analysis. To the beginning, let us derive the explicit virtual constraints from the $i-1$ layer at the reconstruction of the $i$ layer by assuming the activation function is an identity function. The weight constraints of the $i-1$ layer can be expressed as:

$$\mathbf{W}\mathbf{x}_{i-1} = \mathbf{z};$$

Split $\mathbf{W}, \mathbf{z}$ into two parts coherently, i.e.:

$$\begin{bmatrix} \mathbf{W}_+ \\ \mathbf{W}_- \end{bmatrix} \mathbf{x}_{i-1} = \begin{bmatrix} \mathbf{z}_+ \\ \mathbf{z}_- \end{bmatrix} \tag{19}$$

Assume the upper part of the weights $\mathbf{W}_+$ is already full rank, therefore:

$$\mathbf{z}_+ = \mathbf{I}_+\mathbf{z} \tag{20}$$

$$\mathbf{x}_{i-1} = \mathbf{W}_+^{-1}\mathbf{I}_+\mathbf{z} \tag{21}$$

$$\mathbf{z}_- = \mathbf{I}_-\mathbf{z} \tag{22}$$

$$\mathbf{W}_-\mathbf{x}_{i-1} = \mathbf{I}_-\mathbf{z} \tag{23}$$

Substituting Equation 21 into Equation 23, we can derive the following constraints over $\mathbf{z}$ after rearranging:

$$(\mathbf{W}_-\mathbf{W}_+^{-1}\mathbf{I}_+ - \mathbf{I}_-)\mathbf{z} = \mathbf{0} \tag{24}$$

Since the activation function is the identity function, i.e. $\mathbf{z} = \mathbf{x}_i$, the virtual constraints $\mathcal{V}$ that the $i$-th layer has inherited from the weight constraints of $i-1$ layer are:

$$\mathcal{V}\mathbf{x}_i = \mathbf{0}; \ \mathcal{V} = \mathbf{W}_-\mathbf{W}_+^{-1}\mathbf{I}_+ - \mathbf{I}_- \tag{25}$$

Virtual constraints as external constraints are able to compensate the local rank-deficiency of an intermediate layer. For other strictly monotonic activation function like Leaky ReLU, Sigmoid, Tanh, the virtual constraints over $\mathbf{x}_i$ can be expressed as:

$$\mathcal{V}\sigma_{i-1}^{-1}(\mathbf{x}_i) = \mathbf{0} \tag{26}$$

This is not a linear equation system w.r.t. $\mathbf{x}_i$, therefore it is hard to be incorporated in R-GAP. In terms of ReLU the virtual constraints could become further more complicated which will reduce its efficacy. Nevertheless, the reconstruction of the $i$-th layer must take the virtual constraints into account. Otherwise, it will trigger a non-negligible reconstruction error later on. From this perspective, we can see that iterative algorithms like optimization-based attacks can inherently utilize such virtual constraints, which is a strength of O-GAP.

We would like to point out that theoretically the gradient constraints also have the same effect as the weight constraints in the virtual constraints but in a more sophisticated way. Empirical results show that the gradient constraints of previous layers do not have an evident impact on the subsequent layer in the O-GAP, so we have not taken it into account. The number of virtual constraints at $i$-th layer can therefore be approximated by $\sum_{n=1}^{i-1} max(|\mathbf{z}_n| - |\mathbf{x}_n|, 0) - max(|\mathbf{x}_n| - |\mathbf{z}_n| - |\mathbf{W}_n|, 0)$.

## D  RANK ANALYSIS OF THE SKIP CONNECTION

If the skip connection skips one layer, for simplicity assuming the activation function is the identity function, then the layer can be expressed as:

$$f = \mathbf{W}^*\mathbf{x}; \ \mathbf{W}^* = \mathbf{W} + \mathbf{I} \tag{27}$$

where $f$ is the output of this layer, the weight matrix $\mathbf{W}^*$ is clear and the number of weight constraints is equal to $|f|$. While the expression of gradients are the same as without skip connection, since:

$$\nabla\mathbf{W}^* = \nabla\mathbf{W} \tag{28}$$

Therefore the number of gradient constraints is equal to $|\mathbf{W}|$. In other words, without consideration of the virtual constraints, if $|f| + |\mathbf{W}| < |x|$ this layer is locally rank-deficient, otherwise it is full rank. This is the same as removing the skip connection.

If the skip connection skips over two layers, for simplicity assuming the activation function is identity function, then the residual block can be expressed as:

$$\mathbf{x}_2 = \mathbf{W}_1\mathbf{x}_1; \ f = \mathbf{W}_2\mathbf{x}_2 + \mathbf{x}_1 \tag{29}$$

Whereas, the residual block has its equivalent fully connected format, i.e.:

$$\mathbf{W}_1^* = \begin{bmatrix} \mathbf{W}_1 \\ \mathbf{I} \end{bmatrix}; \ \mathbf{W}_2^* = [\mathbf{W}_2 \quad \mathbf{I}] \tag{30}$$

$$\mathbf{x}_2^* = \mathbf{W}_1^*\mathbf{x}_1 = \begin{bmatrix} \mathbf{W}_1\mathbf{x}_1 \\ \mathbf{x}_1 \end{bmatrix} \tag{31}$$

$$f = \mathbf{W}_2^*\mathbf{W}_1^*\mathbf{x}_1 \tag{32}$$

From the perspective of a recursive reconstruction, $f$ is clear, so after the reconstruction of $\mathbf{x}_2$, the input of this block $\mathbf{x}_1$ can be directly calculated by subtracting $\mathbf{W}_2\mathbf{x}_2$ from $f$ as shown in Equation 29. Back to the Equation 31 that means only $\mathbf{x}_2^*$ needs to be recovered. Similar to the analysis for one layer, in terms of the reconstruction of $\mathbf{x}_2^*$, the number of weight constraints is $|f|$ and the number of gradient constraints is $|\mathbf{W}_2|$. On top of that the upper part and lower part of $\mathbf{x}_2^*$ are related, which actually represents the virtual constraints from the first layer. Taking these into account, there are $|\mathbf{W}_2| + |f| + |\mathbf{x}_2|$ constraints for the reconstruction of $\mathbf{x}_2^*$. However, $\mathbf{x}_2^*$ is also augmented compared with $\mathbf{x}_2$ and the number of entries is $|\mathbf{x}_1| + |\mathbf{x}_2|$. To conclude, if $|f| + |\mathbf{W}_2| < |\mathbf{x}_1|$ the residual block is locally rank-deficient, otherwise it is full rank. Seemingly, the constraints of the last layer have been used to reconstruct the input of the residual block due to the skip connection[2]. This is an interesting trait, because the skip connection is able to make the rank-deficient layers like bottlenecks again full rank, as shown in Figure 6. It is worth noticing that the bottlenecks have been commonly used for residual blocks. Further, downsampling residual blocks also have this characteristic of rank condition, as the gradient constraints in the last layer are much more than the first layer due to the number of channels.

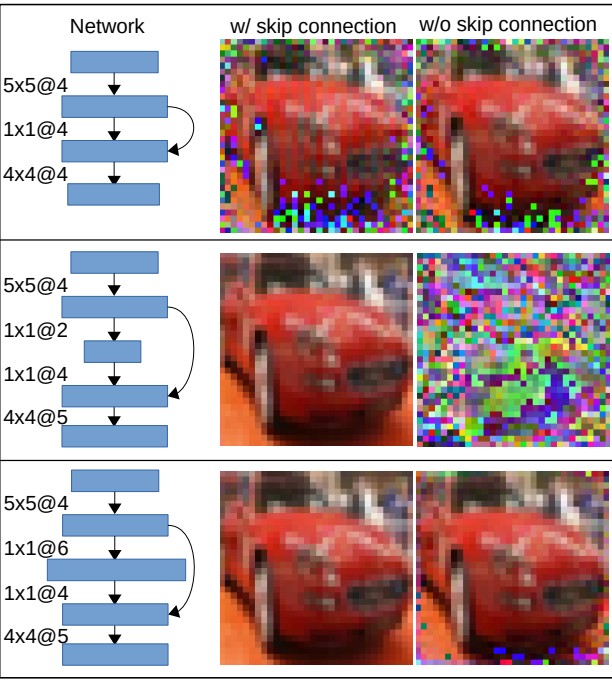

Figure 6: Comparison of optimization-based gradient attacks over architectures with or without the skip connection. The width of blue bars represents the number of features at each layer. The first row shows that there is no impact on the reconstruction if the skip connection skips one layer. The second row shows if the skip connection skips a bottleneck block, which is rank-deficient, the resulting network can still be full rank and enable full recovery of the data. The third row shows the reconstructions of two full-rank architectures. Since the skip connection aids in the optimization process, the quality of its reconstruction is marginally better.

# E IMPROVING DEFENDABILITY OF RESNET101

We also apply the rank analysis to ResNet101 and try to improve its defendability. However, we find that this network is too redundant. It is not possible to decrease the RA-i by cutting a single skip connection as was done in Figure 4. Nevertheless, we devise two variants, the first of which cuts the skip connection of the third residual block and generates a layer that is locally rank-deficient

---

[2]Through formulating the residual block with its equivalent sequential structure, this conclusion readily generalizes to residual blocks with three layers.

and requires a large number of virtual constraints. Additionally, we devise a second variant, which cuts the skip connection of the first residual block and reduces the redundancy of two layers. The accuracy and reconstruction error of these networks can be found in Table 4.

|  | RA-i | Accuracy on val. | MSE of reconstructions |
|---|---|---|---|
| ResNet101 | $-1.4 \times 10^4$ | 91.04% | $0.96 \pm 0.091$ |
| Variant 1 | $-1.4 \times 10^4$ | 90.36% | $1.8 \pm 0.14$ |
| Variant 2 | $-1.4 \times 10^4$ | 90.16% | $1.3 \pm 0.14$ |

Table 4: Training 200 epochs on CIFAR10 and saving the model with the best performance on the validation set, ResNet101 with base width 16 and its two variants achieve similar accuracy on the classification task. The two modified variants which are designed to introduce rank deficiency perform almost as well as the original, but better protect the training data. We conduct a gradient attack with the state-of-the-art approach proposed by Geiping et al. (2020). MSE of the reconstructions of the two rank-deficient variants is significantly higher, which indicates that for deep networks, we can also improve the defendability by decreasing local redundancy or even making layers locally rank-deficient.

## F  R-GAP IN THE BATCH SETTING RETURNS A LINEAR COMBINATION OF TRAINING IMAGES

It can be verified straightforwardly that R-GAP in the batch setting will return a linear combination of the training data. This is due to the fact that in the batch setting the gradients are simply accumulated. The weighting coefficients of the data in this linear mixture are dependent on the various values of $\mu$ for the different training data (see Figure 1). Figures 7 and 8 illustrate the results vs. batch DLG (Zhu et al., 2019) on examples from MNIST.

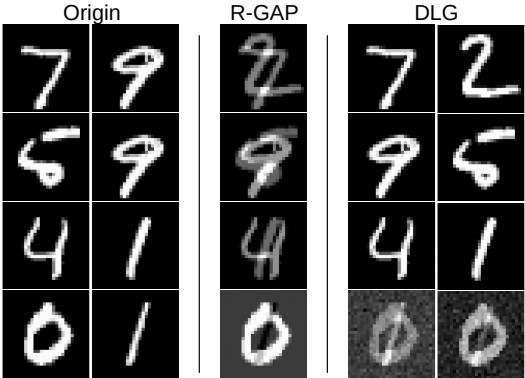

Figure 7: Reconstruction over a FCN3 network with batch-size equal to 2. For FCN network, R-GAP is able to reconstruct sort of a linear combination of the input images. DLG will also works perfectly on such architecture.

## G  ADDING NOISE TO THE GRADIENTS

The effect on reconstruction of adding noise to the gradients is illustrated in Figure 9.

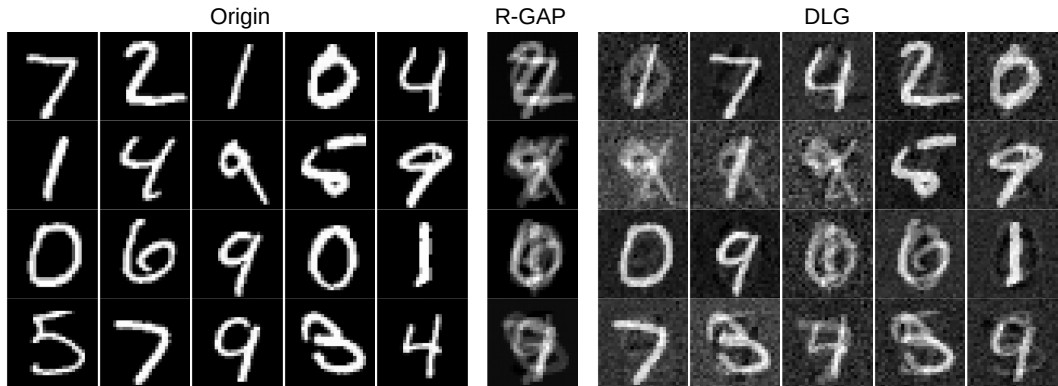

Figure 8: Reconstruction over a FCN3 network with batch-size equal to 5. Sometimes DLG will converge to a image similar to the one reconstructed by the R-GAP.

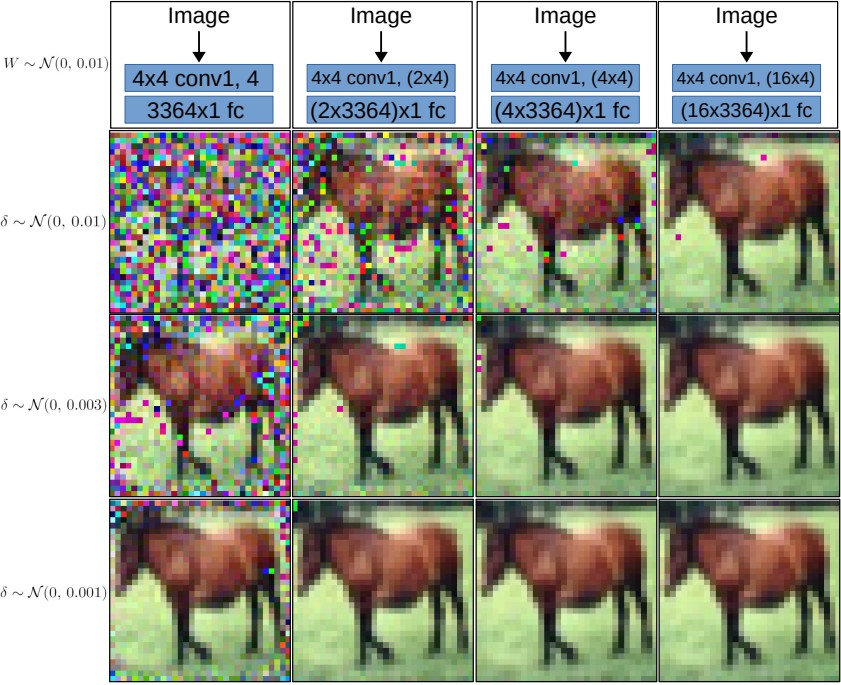

Figure 9: In terms of least square as what we have used for R-GAP, overall increasing the width of the network will involve more constraints and hence enhance the denoising ability of the gradient attack. For O-GAP this also means a more stable optimization process and less noise in the reconstructed image, which has been empirically proven by Geiping et al. (2020). Increasing the width of every layer will definitely decrease the RA-i, so the quality of reconstruction has been improved. Whereas, increasing the width of some layers may not change RA-i of a network, since the RA-i of a network is equal to the largest RA-i among all the layers, i,e, the reconstruction will not get better. However, it is widely believed that more parameter means less secure.

# H   DERIVING GRADIENTS

$$\mu = y\mathbf{w}_d\,\sigma_{d-1}\overbrace{\left(\mathbf{W}_{d-1}\underbrace{\sigma_{d-2}\left(\mathbf{W}_{d-2}\phi\left(\mathbf{x}\right)\right)}_{=:f_{d-2}(\mathbf{x})}\right)}^{=:f_{d-1}(\mathbf{x})} \tag{33}$$

$$\ell = \log(1 + e^{-\mu}) \tag{34}$$

$$d\ell = \frac{-\mu}{1 + e^{\mu}}d\mu; \quad \frac{\partial\ell}{\partial\mu} = \frac{-\mu}{1 + e^{\mu}} \tag{35}$$

$$d\ell = (\frac{\partial\ell}{\partial\mu}y) \cdot d(\mathbf{w}_d f_{d-1}(\mathbf{x})) \tag{36}$$

$$d\ell = (\frac{\partial\ell}{\partial\mu}y) \cdot (d(\mathbf{w}_d)f_{d-1}(\mathbf{x}) + \mathbf{w}_d d(f_{d-1}(\mathbf{x}))) \tag{37}$$

$$d\ell = \frac{\partial\ell}{\partial\mu}y f_{d-1}^{\top}(\mathbf{x}) \cdot d\mathbf{w}_d + (\mathbf{w}_d^{\top}(\frac{\partial\ell}{\partial\mu}y)) \cdot df_{d-1}(\mathbf{x}) \tag{38}$$

$$\frac{\partial\ell}{\partial\mathbf{w}_d} = \frac{\partial\ell}{\partial\mu}y f_{d-1}^{\top} \tag{39}$$

$$d\ell = \frac{\partial\ell}{\partial\mathbf{w}_d} \cdot d\mathbf{w}_d + (\mathbf{w}_d^{\top}(\frac{\partial\ell}{\partial\mu}y)) \cdot (\sigma_{d-1}' \odot df_{d-1}(\mathbf{x})) \tag{40}$$

$$d\ell = \frac{\partial\ell}{\partial\mathbf{w}_d} \cdot d\mathbf{w}_d + ((\mathbf{w}_d^{\top}(\frac{\partial\ell}{\partial\mu}y)) \odot \sigma_{d-1}') \cdot df_{d-1}(\mathbf{x}) \tag{41}$$

$$d\ell = \frac{\partial\ell}{\partial\mathbf{w}_d} \cdot d\mathbf{w}_d + ((\mathbf{w}_d^{\top}(\frac{\partial\ell}{\partial\mu}y)) \odot \sigma_{d-1}') \cdot (d(\mathbf{W}_{d-1})f_{d-2}(\mathbf{x}) + \mathbf{W}_{d-1}d(f_{d-2}(\mathbf{x}))) \tag{42}$$

$$\frac{\partial\ell}{\partial\mathbf{W}_{d-1}} = \left(\left(\mathbf{w}_d^{\top}\left(\frac{\partial\ell}{\partial\mu}y\right)\right) \odot \sigma_{d-1}'\right) f_{d-2}^{\top} \tag{43}$$

$$d\ell = \frac{\partial\ell}{\partial\mathbf{w}_d} \cdot d\mathbf{w}_d + \frac{\partial\ell}{\partial\mathbf{W}_{d-1}} \cdot d\mathbf{W}_{d-1} + \mathbf{W}_{d-1}^{\top}((\mathbf{w}_d^{\top}(\frac{\partial\ell}{\partial\mu}y)) \odot \sigma_{d-1}') \cdot df_{d-2}(\mathbf{x}) \tag{44}$$

$$\cdots$$

