# OpenReview forum: "R-GAP: Recursive Gradient Attack on Privacy"
_ICLR.cc/2021/Conference — ICLR 2021 Poster_

### Official Review · AnonReviewer4 · 2020-10-28
**This paper proposes a new gradient based attack which is able to reconstruct a data point when the gradient with respect to that data point is available**

**Rating:** 7
**Confidence:** 4

**Review:**

This paper studies the problem of gradient attack in deep learning models. In particular,  this paper tries to form a system of linear equations to find a training data point when the gradient of the deep learning model with respect to that data point is available. The algorithm for finding the data point is called R-GAP.

Strengths:
1. The idea of the paper is simple and interesting. The main idea is that for a deep learning model with $d$ layers, we can form $d$ systems of linear equations that can be solved recursively. Solving the last system of linear equations gives us the reconstructed data point.
2. Moreover, because R-GAP is using systems of linear equations, it is very easy to know/analyze the feasibility of the R-GAP algorithm.
3. In contrast to (Phong et al. (2018)), the proposed method works for CNNs as well.

Weaknesses:
1. The contribution of the paper seems limited. This paper extends the method of (Phong et al. (2018)). More precisely, (Phong et al. (2018)) forms a system of equations only with respect to the first layer. The current paper does the same thing for $d$ layers.

2. This paper compares R-GAP with the DLG algorithm. However, there is an improved version of the DLG algorithm called iDLG (Zhao et al. (2020)). Therefore, more numerical examples are required to compare R-GAP with iDLG.
3. This paper situates itself in the Federated Learning (FL) literature. However, in FL problems, each node reports the gradient with respect to the whole local training data points (not a single data point). Therefore, it is not realistic to assume that the central node or an adversary has access to the gradient with respect to one data point. I believe batch DLG (Zhu et al., 2019) is a better attack method in an FL setting.

Update: I would like to increase the score to 7. The authors have improved the paper and addressed my concerns.

---

> ### Author Response · Authors · 2020-11-12
> **Response to AnonReviewer4**
>
> 1. R-GAP is far from a straightforward generalization of the bias attack (Phong et al., 2018) to $d$ layers. The bias attack works only on a fully connected network with bias terms. Therefore, understanding the bias attack cannot help us understand a gradient attack on a convolutional network. R-GAP utilizes much more information and is the first closed-form algorithm that works on both convolutional networks and fully connected networks with or without bias term. Through the study of R-GAP, we have provided many insights into gradient attacks, e.g. the architecture rather than the number of parameters is critical to gradient attacks, why DLG is sensitive to the initialization, why gradient attacks are robust to zero mean noise added to the gradients (this noise is averaged out over redundant estimates of the data either explicitly in the case of R-GAP or implicitly in the case of DLG).
>
> 2. This point was also raised by AnonReviewer3. iDLG proposed a way to analytically derive the label rather than let DLG jointly reconstruct the label and data, which increases the stability of DLG. In our work, as the label is analytically retrievable, we always optimize DLG with the label fixed and let it recover only the image. Thus all of our comparisons are to iDLG and not to the original DLG approach. We will clarify this in a revision of the paper.
>
> 3. In the batch setting, R-GAP in unmodified format returns a linear combination of training data.  This is already a violation of privacy.  A closed-form attack that recovers individual batch samples is an obvious target for future work, and will provide additional insights into the performance of DLG and in which cases recovery of individual samples is feasible.  We empirically observe that DLG has a tendency to mix samples in the batch setting (Figure 7), and it can be of interest to understand what causes this behavior. We do not claim that R-GAP beats DLG in all settings. DLG is a good optimization-based method but with comparatively limited theoretical insights. Through R-GAP we are able to provide many insights about gradient attacks, including DLG. We regard our work as a theory-oriented study, focused on providing insights into how gradient attacks work so that more secure federated learning algorithms can be developed.

---

### Official Review · AnonReviewer2 · 2020-10-29
**This paper addresses a distributed and federated learning based gradient attack techniques, which explains how gradients can lead to recovery of original data.**

**Rating:** 6
**Confidence:** 2

**Review:**

The authors proposed a theoretical explanation of the gradient attack on privacy. To be specific, a closed-form approach, namely recursive gradient attack on privacy has been proposed to explain how the gradients can be used to recover the original data. Moreover, the authors also presented a rank analysis method that is used to estimate the risk of being gradient attacked. Overall, this paper is easy to follow and well-written. I have a few concerns as follows:
1. In Eqn.10, the authors assume the activation function is ReLU or LeakyReLU. In that case, the recursive form will be held. I guess whether the limitation for this method would be the activation function has to be ReLU or LeakyReLU? For instance, sigmoid and tanh activation functions are also widely used. Whether the proposed method is able to address those? If not, what would the reconstruction errors be.

2. I am a little bit confused by the equations (19) and (20). When the activation functions are non-linear, how the authors define V? Also, how does V affect the layer rank? Maybe the conclusion below equation (20) is not very straightforward to me.

3. As for the networks that used in this paper, I wonder when residual connections are used, how the rank deficiency problem would be?

4. The follow-up question is when the network is very deep, like DenseNet and ResNet, whether the conclusions of this paper will hold?

5. The symbols in Algorithm 1 should be consistent to the context. More importantly, I would suggest the authors highlight the correspondences between the equation and the algorithm diagram.

6. Practical usage is one of my concern. In practice, researchers or engineers may use very deep models for practical problems. If the authors could demonstrate a more practical model, that would make this submission very strong.

---

> ### Author Response · Authors · 2020-11-12
> **Response to AnonReviewer2**
>
> 1. Eq. (10) shows that if the activation function is LeakyReLU or ReLU we can average different estimates of $\frac{\partial \ell}{\partial \mu}\mu$ computed at different layers in order to obtain a more robust estimate. If not, we can still use the result of the last layer as shown in Eq. (9).
>
> 2. We will revise our explanation of the virtual constraints in the revision to increase clarity. When the activation function becomes non-linear, so do the virtual constraints. For a strictly monotonic activation function (which holds for most common choices), the virtual constraints can be expressed as $\mathcal{V}\sigma^{-1}(\textbf{x}) = 0$.  We can still count the number of such constraints to provide an estimate of the data vulnerability.
>
> 3. This is a very good question. The residual connection has some interesting traits in terms of the rank deficiency: if the section of the network it skips over is rank deficient, the resulting network can still be full rank and enable full recovery of the data. We will add a new subsection in the appendix discussing this, and we have already performed experiments showing the effect.
>
> 4. See also the response to AnonReviewer3 and additional experiment.  Geiping et al. (2020) report that network depth does not tend to protect privacy in the context of DLG.  Although there may be some accumulation of error over the layers with R-GAP, this tends to be minimal for common choices of activation function.  For less stable activation functions (e.g. Sigmoid), we still get a recognizable reconstruction therefore breaking privacy.
>
> 5. The variable names are consistent with preceding equations (9)-(16).  We will note the correspondence in the caption of the image in the revised version.
>
> 6. See response to AnonReviewer3 regarding additional architectures.

---

### Official Review · AnonReviewer3 · 2020-11-08

**Rating:** 7
**Confidence:** 3

**Review:**

This paper proposes a new gradient attack method named R-GAP. R-GAP decomposites the DNN gradient attack problem into subproblems for each layer, and recursively solves each of them. The subproblem of each layer is formulated as a least-square reconstruction problem.

The authors further point out the rank of network weight matrix is (non-surprisingly) correlated with the difficulty of input recovery. Based on this finding, they design a metric based on the weight matrix rank to estimate the feasibility of fully recovering data.
Experimental results on MNIST and CIFAR10 show that the proposed method R-GAP is comparable or superior to the classic DLG method. The authors also claimed the proposed method to be much faster than DLG baseline.

Strengths:
1. Gradient attacks raise data privacy concerns in many applications such as federated learning, making it an important problem.
2. As an analytical method to solve the input reversion problem, R-GAP should have its intrinsic advantage over previous optimization-based gradient attack (O-GAP) methods such as DLG. For example, in my assumption (and also claimed by the authors), R-GAP can be much faster than O-GAP.
3. R-GAP has much better performance than DLG on full-rank CNN6 networks, as shown both visually and numerically in Figure 3 and Table 1. This shows that if the attacked model satisfies the full-rank condition, R-GAP can be both faster and more effective than DLG.

Weakness:
1. Insufficient experiments.
My largest concern is over the lack of necessary experiments to show the advantage of R-GAP.
i. Why only showing results on a self-designed CNN6 network? In order to fairly compare with DLG, and also to show the general effectiveness of the proposed method, the authors should also consider comparing with DLG on some standard network, such as the LeNet benchmarked in many previous gradient attack works [1,2].
This is very important also because we need to see whether the popular deep models such as LeNet, VGG, ResNet, etc. satisfy the full rank condition required by the proposed method. If not, the proposed R-GAP will have limited application scenarios.
ii. Why not compare with more recent gradient attack methods such as iDLG [2], which has been shown to also outperform the original DLG [1]?
iii. The authors claimed the proposed method is much faster than DLG. Although I agree this is intuitively true, I think it necessary to report the numbers in the paper. For example, how much time/FLOPs does it take to attack a single image for each method?

2. Additional tricks used without detailed description in the Method section.
In Table 1, the authors show that R-GAP is largely outperformed by DLG on the rank-deficient network CNN6-d. However, according to the authors' vague descriptions, simply adding a smoothing operation can largely improve R-GAP performance. (I assume H-GAP = R-GAP + image smoothing?) Is the image smoothing the main technique making the proposed method effective?
This is really confusing since neither H-GAP nor image smoothing is mentioned in the method/related work sections. I suggest the authors to provide more descriptions about the H-GAP method and also provide explanations why it largely outperforms R-GAP.

I'm willing to increase my score if these concerns are properly tackled during the rebuttal period.

Other comments:
1.  The artifacts of DLG reconstruction images are mainly located on the corner of the images (see Figure 2), while the artifacts of DLG are evenly distributed on the whole images (see Figure 3). Is this a general trend? If yes, any explanations or intuitions behind this?
2. The proposed RA-i is only using matrix rank to predict the hardness of input recovery. In my view, it might be better to consider using matrix condition number as the metric. This is because the sub-problem at each layer is basically a least-square regression problem, and two least-square regression problems can have different difficulties when the matrix have identical ranks but different condition numbers. In other words, condition number contains more information than rank, and thus might be more useful. (Please point out if I'm wrong here.)

Reference

[1] Deep leakage from gradients.

[2] iDLG: Improved Deep Leakage from Gradients.

Update: The authors have addressed my concerns and now I vote for acceptance.

---

> ### Author Response · Authors · 2020-11-12
> **Response to AnonReviewer3 comment 1**
>
> W1: (i) The rank constraints are not only relevant to R-GAP, but to any gradient attack method.  Our rank analysis also predicts when the *optimization attack* (DLG) fails. We have identified three types of constraints in the gradient attack problem: gradient constraints, weight constraints and virtual constraints. The first two constraints are linear and can be easily recursively defined, and therefore have been taken into account by R-GAP, which regards the attack as a sequence of linear systems. Whereas, DLG is an optimization-based attack which implicitly takes all of them into account. However, R-GAP is a closed-form algorithm which is amenable to analysis and is not susceptible to local optima. As such, we should not view R-GAP as a direct competitor to DLG.  Indeed, we may consider a possible application of R-GAP as an initialization of DLG (see additional experiment in the next paragraph), although in our experiments R-GAP usually produces results that break privacy by itself.
>
> LeNet also satisfies the rank conditions for full recovery of the training data with R-GAP.  Our code release will include the calculation of the rank conditions, and we will include additional analysis for popular network architectures in the final version of the paper. The following table summarizes the results of an additional experiment on the LeNet architecture (evaluated on CIFAR-10).
>
> Condition Number:
>
> Network|      conv1        |       conv2       |        conv3     |
>
> LeNet | $1.8\times 10^4 \pm 29.1$            | $6.1 \times 10^3 \pm 3.1$             | $32.4 \pm 2.9 \times 10^{-3}$ |
>
> LeNet*| $1.2\times 10^3 \pm 2.0\times 10^2$ | $1.3\times 10^3 \pm 2.3\times 10^2$ | $14.2 \pm 0.45$             |
>
> MSE:
>
> Network|         DLG         |       R-GAP      | R-GAP-> DLG |
>
> LeNet  |   $3.7\times 10^{-8} \pm 8.6\times 10^{-9}$  |  $1.1\times 10^{-4} \pm 7.8 \times 10^{-5}$  |   $1.1 \times10^{-6} \pm1.1 \times 10^{-5}$  |
>
> LeNet*|   $5.2\times 10^{-2}\pm 2.9\times10^{-2}$  | $1.5\times 10^{-10}\pm2.5\times10^{-10}$ |    $4.8\times10^{-4}\pm 9.1 \times 10^{-4}$  |
>
> LeNet* is identical to LeNet but uses a Leaky ReLU activation function instead of Sigmoid
>
>
> Both DLG and R-GAP perform well on LeNet. Empirically, if the MSE is around or below $1\times10^{-4}$, the difference of the reconstruction will be visually undetectable. However, we surprisingly find that by replacing the Sigmoid function with the Leaky ReLU, the reconstruction of DLG becomes much poorer. The condition number of the matrix $\textbf{A}$ (from Algorithm 1 in the paper) changes significantly in this case. Since the Sigmoid function leads to a higher condition number at each layer, any reconstruction error in the subsequent layer will be amplified in the previous layer, therefore DLG is forced to converge to a better result. In contrast, R-GAP has an accumulated error and naturally performs much better on LeNet*. Additionally, we find R-GAP could be a good initiliazation tool for DLG. By initiliazing DLG with the reconstruction of R-GAP, and running 8\% of the previous iterations, we achieve a visually indistinguishable result.  However, for LeNet*, we find that DLG reduces the reconstruction quality obtained by R-GAP, which further shows the instability of DLG.
>
> (ii) "Improved Deep Leakage from Gradients" (iDLG) proposed a way to analytically derive the label rather than let DLG jointly reconstruct the label and data. In our work, as the label can be analytically recovered, we are always providing DLG the ground-truth label and let it recover the image only. Therefore the experiment is actually comparing R-GAP with iDLG, we will add a note explaining this to the paper.
>
> (iii) The cost of running R-GAP is approximately the same order of magnitude as a single iteration of DLG.  A step of DLG is dominated by computing the gradient of the gradient w.r.t. the function input, while R-GAP is dominated by the matrix inversions at each step (cubic in the number of rows).  We note that both methods can run on a GPU.  The difference in run-time between the methods is orders of magnitude (not even close), and is dominated by the large number of optimization iterations required by DLG (typically hundreds as reported in their paper).

---

> > ### Author Response · Authors · 2020-11-12
> > **Response to AnonReviewer3 part 2**
> >
> > * W2: H-GAP is simply a hybrid attack that selects the better output of either R-GAP or DLG.  The selection is done by rejecting the solution that has more salt-and-pepper type noise.  We measure this by the difference of the image and its smoothed version (achieved by a simple 3x3 averaging) and select the output with the smaller norm.  In this way, we can always select the better of the two outputs by taking into account a minimal amount of domain knowledge.  We note that other recent work has extended DLG to incorporate image priors, and our approach is a simple step in this vein.  We will improve the description in the paper.
> >
> > * C1: Figure 2 shows reconstruction artifacts when the network is rank deficient, while Figure 3 shows artifacts caused by convergence to a local minimum in the optimization process. For the rank deficient case, the convolution kernel performs less computation at the edges and corners, and therefore that is where the artifacts are located.
> >
> > * C2: We have included an analysis of the condition number in the above experiment.  We note that the condition number is data and network dependent. By contrast, rank analysis is an offline tool, which can be useful to understand risks inherent in certain network architectures.  Ultimately, it is not our goal to attack networks, but to show via these attacks when data are vulnerable (regardless of whether the optimization is hard or easy).  Our goal is also to provide a basis for analysis that will ultimately lead to a better understanding of how to design data-secure federated learning systems.
> >
> > Back to the goal of this work, DLG was published in NeurIPS 2019 last December.  Although it has received a bit less than 100 citations in less than a year, until now there is little theoretical understanding of it. Through our study, we found that a gradient attack is similar to solving a chain of linear equations and proposed the R-GAP based on this idea. R-GAP is more stable and consumes less time than the optimization-based attack, but does not take the non-linear constraints into account. However, R-GAP is a closed-form algorithm and easy to study. Therefore, we are able to provide insights on many important open questions of DLG, e.g. why DLG is sensitive to its initialization, local optima due to twin-data, how to estimate the risk of gradient attacks based on a network architecture, etc.

---

### Author Response · Authors · 2020-11-17
**Revision and generic comment**

We thank the reviewers for their helpful comments and insightful reviews. With your feedback, we are able to provide better work.

In this revision we have added:
* an experiment comparing R-GAP with DLG over LeNet. We further provide some insights of the instability of DLG through the condition number of constraints matrix A.
* a section Appendix D about rank analysis of the skip connection. We analytically prove and empirically demonstrate that the skip connection can make rank-deficient bottle-neck again full rank.
* an experiment about improving the security of ResNet18 using rank analysis. More importantly, we demonstrate that we can improve the defendability towards gradient attacks without sacrificing accuracy.

We note that in our response to AnonReviewer4, we referred to Figure 7 (old numbering).  In the revision, this now refers to Figure 8 (new numbering).

We have also edited:

* section of the rank analysis, to make the explanation of virtual constraints more clear.
* the abstract, introduction and results, conclusion to emphasize that the focus of this work include not only the attack approach but also a risk estimation tool and other theoretical understanding of gradient attacks.

---

> ### Author Response · Authors · 2020-11-24
> **Final revision**
>
> We have uploaded our final revision, in this revision we:
>
> * insert some qualitative results to Figure 4.
> * add a new section Appendix E talking about improving the defendability of ResNet101 according to the rank analysis.
>
> We would like to thank three reviewers for their work and valuable feedback.

---

### Decision · Program_Chairs · 2021-01-07
**Final Decision**

**Decision:**

Accept (Poster)

**Comment:**

The major criticism of this paper after the initial reviews was a lack of experimental results on deeper and more modern architectures that include skip connections.  The authors added results to the paper to address these issues.